# The impact of resident involvement and postgraduate year on head and neck surgery for obstructive sleep apnea

Mihai A. Bentan[1]☯*, Elizabeth Mastoloni[2]☯, Lawrance Lee[1]☯, Ryan Nord[1]☯

1 Department of Otolaryngology–Head and Neck Surgery, Virginia Commonwealth University, Richmond, Virginia, United States of America, 2 Virginia Commonwealth University School of Medicine, Richmond, Virginia, United States of America

☯ These authors contributed equally to this work.
* bentanm@vcu.edu

**Data Availability Statement:** The data is part of the ACS-NSQIP PUF database maintained by the American College of Surgeons. The data is

## Abstract

To assess the impact of resident involvement and resident postgraduate year (PGY) on head and neck obstructive sleep apnea (OSA) surgical outcomes. We analyzed head and neck OSA surgeries from 2005–2012 via the National Surgical Quality Improvement Program database. Demographic, preoperative, and postoperative variables were analyzed via multivariate regression to determine the impact of resident involvement and resident PGY on 30-day outcomes. Of 975 surgeries, 234 (24.0%) involved a resident: 120 (51.3%) involved a junior (PGY 1–3) resident and 114 (48.7%) involved a senior (PGY 4–5) resident. Multivariate analysis showed no significant impact on surgical, medical, or overall complication rates with resident involvement (all p > 0.05). Likewise, after separation of involved residents into junior or senior cohorts based on PGY, neither junior nor senior resident involvement significantly increased the odds of medical, surgical, or overall complications compared to operations performed by the attending alone (all p > 0.05). Resident involvement significantly increased readmission rates (6.1% versus 1.4%, p = 0.041) and operation time (92.1 ± 59.9 minutes versus 53.6 ± 42.0 minutes, p < 0.001) when compared to cases without resident involvement. Resident involvement in OSA surgery does not significantly impact rates of surgical medical, or overall complications. However, resident involvement increases 30-day readmission rates and almost doubles operation time, suggesting that resident involvement in head and neck OSA surgery remains relatively safe but further efforts to improve efficiency are likely needed.

## Introduction

Obstructive sleep apnea (OSA), the partial or complete collapse of the upper airway during sleep resulting in arousal or decreased oxygen saturation, is estimated to affect almost 1 billion people globally [1]. Risk factors include obesity, advancing age, and male sex [1, 2]. When left untreated, OSA increases the risk for hypertension, diabetes, stroke, and negatively impacts

available free of charge from the American College of Surgeons for members of hospitals participating in the NSQIP. Data access queries for non-members can be directed to the Corresponding Author, Mihai Bentan (bentanm@vcu.edu). Data requests may also be performed via the following email address: acsnsqip@facs.org. Public disposition of this data would breach the Data Use Agreement (and therefore copyright laws) as imposed by the American College of Surgeons, as detailed in the Participant Use Data File 'Access to Chapter 4' section: "Chapter 4 (for both Essential and Procedure Targeted data) contains proprietary ACS NSQIP programmatic information. Your use of Chapter 4 is restricted to clarifying definitions for your PUF-based research purposes. The contents of Chapter 4 shall not be shared, quoted, distributed, or disseminated in any published papers, reports, manuscripts, etc. Limited paraphrasing of Chapter 4 definitions, to provide methodological clarity in research publications, is permissible. Any use or redistribution of Chapter 4 for commercial purposes is strictly prohibited." Therefore, given these copyright laws, we are unable to provide the data underlying our finals in any of the aforementioned methods (a-c) so as to avoid violating copyright laws. However, individuals/readers interested seeking permission to use the data may follow contact the ACS who will process a request, as detailed in the following: "Permission to use and disclose the PUF is granted from the ACS NSQIP to each Data Recipient. The Data Recipient must be an employee of the participating Hospital. Intentional misrepresentation of employment at the Hospital by the Data Recipient will void this Agreement, prohibit use of the PUF by the Data Recipient, and result in legal action taken by the ACS NSQIP. The ACS NSQIP also reserves the right to deny access to the PUF at its discretion." Further information regarding all of this information can be found at the following link: https://www.facs.org/media/ezaj2g5p/datauseagreementacsnsqippuf.pdf The data is owned by a third-part institution (ACS NSQIP) and therefore they cannot be shared by any of the authors of this study. The following URL: https://www.facs.org/quality-programs/data-and-registries/pediatric/participant-use-data-file/participant-use-request-form/ should be utilized to identify the appropriate non-author contact for the location/state in which the requesting individual resides. Other individuals are able to access this data in the same manner that these authors have. We report that these authors did not have any specific access privileges that others would not have had. Data requests may be performed using

mental health and quality of life [1–3]. Continuous positive airway pressure (CPAP) therapy remains the first-line treatment for OSA [1]. But, when CPAP or other non-surgical interventions have failed, often due to poor patient adherence [4, 5], surgical intervention may be warranted in select candidates [6–8].

Effective surgical management of OSA demands specialized skills and training, highlighting the crucial role of otolaryngology surgical residency in providing the necessary education and competencies for performing these operations safely. Nonetheless, the hands-on surgical experience for residents has decreased following the implementation of an 80-hour weekly work limit by Accreditation Council for Graduate Medical Education (ACGME) in 2003. This change has led to surgical residents taking on less significant roles in surgical cases, correlating with an increase in complication rates while worsening patient outcomes at these teaching hospitals [9]. Considering these challenges, teaching hospitals must balance the imperative of training the next generation of surgeons and optimizing patient safety.

Previous research has examined the influence of surgical residents on procedure outcomes [10–13], but prior data has not specifically focused on how their involvement affects patient outcomes in head and neck OSA surgery. Given the rising incidence of OSA [14], further research on this topic is warranted. Numerous studies have utilized the National Surgical Quality Improvement Program (NSQIP), a database developed by the American College of Surgeons (ACS), to assess 30-day postoperative outcomes of surgical procedures [10–13, 15–23]. However, to date, no studies have utilized the ACS NSQIP database to investigate the effects of resident involvement on OSA surgical outcomes. This study aims to explore how the involvement of residents, along with their postgraduate year (PGY), influences postoperative outcomes in patients undergoing head and neck surgery for OSA.

## Materials and methods

A retrospective analysis utilizing patients who underwent head and neck surgery for OSA from 2005 to 2012 was performed using the ACS NSQIP database. A list of all surgical procedures performed can be found in Table 1. Similar to the approach outlined by Knoedler et al. [24], only cases containing the international classification diagnosis 9th edition (ICD-9) code 327.33 or international classification diagnosis 10th edition (ICD-10) code G47.33 were included. Surgeries describing non-head and neck operations (e.g. bariatric surgery) were excluded from this study. Cases without data on resident involvement or missing post-graduate year (PGY) of involved residents were also excluded. Likewise, all cases detailing fellow (PGY6+) involvement were also excluded from analysis, given their low sample size (n = 15) and the variability in non-standardized fellowship instruction that may involve surgical training, medical management, or a hybrid experience.

Demographic characteristics collected included age, sex, race, and body mass index (BMI). Preoperative characteristics collected included diabetes, smoking status, dyspnea, alcohol use, pneumonia present at the time of surgery, significant weight loss >10% of body weight within 6 months prior to surgery, chronic obstructive pulmonary disease (COPD), congestive heart failure (CHF), ventilator dependence, dialysis dependence, previous myocardial infarction (MI), previous percutaneous coronary intervention (PCI), previous cardiac surgery (e.g. coronary artery bypass grafting, valve replacement or repair, cardiac transplant), history of angina within one month prior to surgery, hypertension requiring medication, history of transient ischemic attack (TIA), history of cerebrovascular accident (CVA) with neurologic deficit, history of CVA without neurologic deficit, disseminated cancer, steroid use for chronic condition, bleeding disorder (e.g. vitamin K deficiency, hemophilia, but *not* including those who are on chronic aspirin therapy), systemic sepsis present at the time of surgery, wound infection at

the aforementioned URL link or via the following email address: acsnsqip@facs.org.

**Funding:** The author(s) received no specific funding for this work.

**Competing interests:** I have read the journal's policy and the authors of this manuscript have the following competing interests: Dr. Ryan Nord is a consultant for Inspire Medical, Nyxoah, and SIESTA medical. All other authors have declared that no competing interests exist.

**Table 1. Included current procedural terminology codes with patient distribution.**

| Procedure | N |
|---|---|
| Isolated PPP | 501 |
| + Septoplasty | 4 |
| + Septoplasty + Tongue Ablation (Radiofrequency) | 1 |
| + Septoplasty + Uvulectomy | 1 |
| + Tongue Ablation (Radiofrequency) | 8 |
| + Hyoid myotomy with suspension | 2 |
| + partial glossectomy | 1 |
| + tongue base suspension permanent suture | 1 |
| Isolated PPP + T/A | 196 |
| + Septoplasty | 9 |
| + Septoplasty + Sinus Surgery | 1 |
| + Sinus Surgery | 1 |
| + Tongue Ablation (Radiofrequency) | 5 |
| + partial glossectomy | 3 |
| Isolated PPP + T/A + Turbinate Reduction | 19 |
| + Septoplasty | 29 |
| + Septoplasty + Sinus Surgery | 3 |
| + Septoplasty + Sinus Surgery + Tongue Ablation (Radiofrequency) | 1 |
| + Septoplasty + Tongue Ablation (Radiofrequency) | 3 |
| + Septoplasty + partial glossectomy | 1 |
| + Sinus Surgery | 2 |
| + Sinus Surgery + Tongue Ablation (Radiofrequency) | 1 |
| + Tongue Ablation (Radiofrequency) | 4 |
| + partial glossectomy | 2 |
| Isolated PPP + Turbinate Reduction | 18 |
| + Septoplasty | 60 |
| + Septoplasty + Sinus Surgery | 11 |
| + Septoplasty + Sinus Surgery + Tongue Ablation (Radiofrequency) | 1 |
| + Septoplasty + Tongue Ablation (Radiofrequency) | 11 |
| + Septoplasty + Hyoid myotomy with suspension | 2 |
| + Sinus Surgery | 3 |
| + Tongue Ablation (Radiofrequency) | 10 |
| + partial glossectomy | 1 |
| Isolated T/A | 88 |
| + Turbinate Reduction | 2 |
| + Turbinate Reduction + Septoplasty | 13 |
| + Turbinate Reduction + Septoplasty + Sinus Surgery | 1 |
| + Turbinate Reduction + Septoplasty + Sinus Surgery + Tongue Ablation (Radiofrequency) | 1 |
| + Turbinate Reduction + Septoplasty + Uvulectomy | 3 |
| + Septoplasty | 1 |
| + hyoid myotomy with suspension | 1 |
| + hyoid myotomy with suspension + Tongue base suspension permanent suture | 3 |
| + partial glossectomy | 1 |
| + uvulectomy | 3 |
| Isolated uvulectomy | 8 |
| + turbinate reduction | 1 |
| + turbinate reduction + Septoplasty | 1 |

*(Continued)*

**Table 1.** (Continued)

| Procedure | N |
|---|---|
| + turbinate reduction + Septoplasty + sinus surgery | 1 |
| + Septoplasty | 1 |
| Isolated hyoid myotomy with suspension | 10 |
| + tongue base suspension permanent suture | 1 |
| Isolated partial glossectomy | 3 |

the time of surgery, surgical location (inpatient versus outpatient), functional health status of the patient, and American Society of Anesthesiologists (ASA) classification. An overall work relative value unit (WRVU) was calculated to act as a proxy for case complexity, as demonstrated in the work of Baker et al [12]. This was calculated by summing the WRVUs of each CPT code, including other and concurrent, to provide an overall WRVU.

Postoperative complications included rates of medical complications, surgical complications, overall complications, mortality, readmission, reoperation, total operating time, and length of hospital stay. Surgical complications included superficial surgical site infection (SSI), deep SSI, organ/space SSI, and blood transfusion within 72 hours of surgery. Medical complications included pneumonia, unplanned reintubation, urinary tract infection, deep vein thrombosis, renal insufficiency, pulmonary embolism, ventilator use for $> 48$ hours, acute renal failure, CVA with neurologic deficit, cardiac arrest requiring cardiopulmonary resuscitation, myocardial infarction (MI), sepsis, and septic shock. Overall complications included medical complications and surgical complications. More information pertaining to all collected demographics, preoperative characteristics, and postoperative complications, can be found in the ACS NSQIP participant use data file guides [25].

Two separate analyses were performed. The first aimed to analyze the impact of resident involvement on postoperative outcomes. Drawing guidance from the approach outlined by Wong et al. in their analysis of resident and fellow participation in transsphenoidal pituitary surgery outcomes [16], cases were separated into two cohorts: those involving residents and those performed by the attending alone. Univariate analyses utilizing t-test and chi-squared test for continuous and categorical variables were performed, respectively. To control for confounding variables, multivariate regression was performed using demographic and preoperative variables significant on univariate analysis. A second, separate analysis was then performed to determine the impact of resident PGY on postoperative outcomes. This was achieved by separating participating residents into subcategories according to their PGY: senior resident (PGY 4–5) or junior resident (PGY 1–3). This subcategorization is similar to the approach outlined by Brady et al. in their analyses of the effect of training level on head and neck free flap surgical outcomes [15]. Univariate analysis utilizing one-way analysis of variance (ANOVA) and chi-squared test for continuous and categorical variables was performed, respectively. Similarly, multivariate regression was utilized to control for confounding demographic and preoperative characteristics significant on univariate analysis. A p-value $< 0.05$ was considered statistically significant. All analyses were performed using JMP, version 17.0.0. SAS Institute Inc. (Cary, NC). Because the ACS NSQIP dataset provides only deidentified patient information, this study did not require Institutional Review Board approval.

## Results

Of the 975 cases that met inclusion criteria, 234 (24.0%) were performed with the assistance of a resident. Of these 234, 120 (51.3%) involved a junior resident whereas 114 (48.7%) involved a

senior resident. Patient demographics and comorbidities comparing cases with or without resident involvement are detailed in Table 2, with comparisons of their postoperative outcomes detailed in Table 3. Rates of readmission (6.1% versus 1.4%, p = 0.041) and total operation time (92.1 ± 59.9 minutes versus 53.6 ± 42.0 minutes, p < 0.001) were significantly increased in cases that involved a resident (Table 3). To control for confounding variables, multivariate analysis was performed using the following variables: race, alcohol use, history of COPD, history of angina, history of CVA with neurologic deficit, and resident involvement. Multivariate analysis demonstrated that resident involvement did not significantly increase the rates of surgical (Odds Ratio [OR] 1.06, 95% Confidence Interval [CI] 0.20–5.73, p = 0.945), medical (OR 1.20, 95% CI 0.30–4.79, p = 0.799), or overall (OR 1.16, 95% CI 0.39–3.43, p = 0.794) complications (Table 4).

With respect to subcategorization of involved residents based on PGY, patient demographics and comorbidities are detailed in Table 5, with comparisons of postoperative outcomes detailed in Table 6. In the multivariate analysis, we controlled for the following variables: race, alcohol use, CHF, COPD, history of previous MI, history of angina, history of CVA with neurologic deficit, significant weight loss, and resident PGY, as separated into one of the three following cohorts: attending alone, senior resident involvement, or junior resident involvement. Rates of readmission were significantly higher in the senior resident cohort (10.3%) compared to the junior resident cohort (3.8%) or the attending alone cohort (1.4%, p = 0.019). Likewise, operation time was significantly higher for the senior resident cohort (90.2 ± 66.1 minutes) and the junior resident cohort (94.0 ± 53.6 minutes) compared to the attending alone cohort (53.6 ± 42.0 minutes, p < 0.001) (Table 6). On multivariate analysis, the senior resident cohort did not significantly increase the odds of surgical (OR 2.01, 95% CI 0.36–11.37, p = 0.429), medical (OR 1.32, 95% CI 0.28–6.12, p = 0.725), or overall (OR 1.62, 95% CI 0.51–5.20, p = 0.417) complications when compared to the attending alone cohort. Similarly, the junior resident cohort did not significantly increase the odds of medical (OR 0.40, 95% CI 0.04–4.07, p = 0.400) or overall (OR 0.23, 95% CI 0.02–2.20, p = 0.205) complications compared to the attending alone cohort. Notably, because no surgical complications were reported in the junior resident cohort, the surgical complications OR could not be determined (Table 7).

## Discussion

The management of OSA through surgical interventions remains indispensable, with multiple studies demonstrating its, albeit variable, efficacy in treating OSA [26, 27]. Sites of airway obstruction differ from one patient to the next, with literature demonstrating varying rates of multi-site obstruction (50.0% - 96.1%) [28–31]. Surgery aimed at improving OSA typically targets multiple sites of obstruction including the nasal airway, velum, oropharynx, tongue base, epiglottis, and more. Nasal obstruction, responsible for up to one-half of total airway resistance, is frequently addressed with septoplasty or turbinate reduction to improve nasal airway patency [32]. While nasal airway surgery alone has rarely been shown to significantly improve the apnea-hypopnea index (AHI) in patients [33], literature has demonstrated nasal airway surgery's success in improving CPAP tolerance and adherence [33–35]. Surgical interventions such as the adenotonsillectomy and uvulopalatopharyngoplasty may help address obstruction of the velum and oropharynx through reshaping the upper airway to ultimately reduce the degree of soft tissue collapse [26, 32, 36, 37]. Surgeries to address soft tissue collapse at the tongue base and epiglottis include glossectomy, tongue base radiofrequency ablation, maxillo-mandibular advancement, and more [38–41]. The implantable hypoglossal nerve stimulator has similarly shown efficacy in improving OSA in select patients [42, 43], but given its approval by the United States Food and Drug Administration in 2014, it was not included in our study.

**Table 2. Overall patient demographics and comorbidities in sleep apnea surgery by resident involvement.**

| | With resident involvement N = 234 | Attending alone N = 741 | P value |
|---|---|---|---|
| Demographics | | | |
| Age, years ± SD | 44.2 ± 12.2 | 43.7 ± 12.0 | .582 |
| Sex, % | | | .090 |
| Female | 31.2 | 25.4 | |
| Male | 68.8 | 74.6 | |
| Race, % | | | < .001* |
| Black | 19.2 | 5.7 | |
| Other | 2.6 | 3.9 | |
| Unknown | 22.2 | 22.4 | |
| White | 56.0 | 68.0 | |
| BMI, kg/m$^2$ ± SD | 32.4 ± 6.8 | 32.9 ± 7.7 | .398 |
| Comorbidities, % | | | |
| Diabetes | 8.1 | 10.4 | .378 |
| Current smoker | 17.1 | 18.5 | .697 |
| Dyspnea | 7.7 | 7.3 | .886 |
| Alcohol | 3.8 | 1.2 | .021* |
| Pneumonia | 0 | 0 | - |
| Weight loss | 0.4 | 0 | .240 |
| COPD | 3.0 | 0.7 | .011* |
| CHF | 0.4 | 0 | .240 |
| Ventilator dependent | 0 | 0.1 | 1 |
| Dialysis dependent | 0 | 0.1 | 1 |
| Previous MI | 0.4 | 0 | .241 |
| Previous PCI | 0.4 | 2.2 | .089 |
| Previous cardiac surgery | 0.9 | 1.5 | .774 |
| Angina | 1.3 | 0.1 | .045* |
| Hypertension | 32.1 | 31.4 | .872 |
| TIA | 0.4 | 0.7 | 1 |
| CVA with neurologic deficit | 1.7 | 0.1 | .013* |
| CVA without neurologic deficit | 1.7 | 0.8 | .264 |
| Disseminated cancer | 0 | 0 | - |
| Steroid use | 0.9 | 0.8 | 1 |
| Bleeding disorders | 0.4 | 0.4 | 1 |
| Systemic sepsis | 0.4 | 0.3 | .565 |
| Wound infection | 0 | 0.3 | 1 |
| Outpatient status | 67.0 | 71.6 | .208 |
| Functional status | | | 1 |
| Dependent | 0.4 | 0.4 | |
| Independent | 99.6 | 99.6 | |
| ASA class | | | .224 |
| 1 | 2.6 | 5.7 | |
| 2 | 62.2 | 59.5 | |
| 3 | 34.3 | 34.4 | |
| 4 | 0.9 | 0.4 | |

*(Continued)*

**Table 2.** (Continued)

| | With resident involvement N = 234 | Attending alone N = 741 | P value |
|---|---|---|---|
| WRVU | 12.5 ± 6.0 | 12.3 ± 6.2 | .670 |

Abbreviations: ASA, American Society of Anesthesiologists classification; BMI, body mass index; COPD, chronic obstructive pulmonary disease; CHF, congestive heart failure; CVA, cerebrovascular accident; MI, myocardial infarction; PCI, percutaneous coronary intervention; TIA, transient ischemic attack; SD, standard deviation; WRVU, work relative value unit

* denotes statistical significance (P < 0.05)

These interventions are typically performed in conjunction with one another to address multiple sites of airway resistance and maximize surgical benefit [44, 45]. However, these surgeries demand high levels of skill and precision from the surgeon. In this context, the involvement of residents in surgical procedures has been a topic of ongoing debate, balancing the imperative role of hands-on training against the paramount goal of patient safety. Our study utilizes the

**Table 3. Postoperative complications in sleep apnea surgery by resident involvement.**

| | With resident involvement N = 234 | Attending alone N = 741 | P value |
|---|---|---|---|
| Overall complications, % | 3.4 | 1.9 | .170 |
| Surgical complications overall, % | 1.3 | 0.8 | .455 |
| Superficial SSI | 0.4 | 0.4 | 1 |
| Deep SSI | 0 | 0 | - |
| Organ/space SSI | 0 | 0.3 | 1 |
| Wound disruption | 0.9 | 0.1 | .145 |
| Blood transfusion within 72 hours | 0 | 0 | - |
| Medical complications overall, % | 2.1 | 1.1 | .208 |
| Pneumonia | 0 | 0.4 | 1 |
| Unplanned reintubation | 0 | 0.7 | .345 |
| Urinary tract infection | 0 | 0.4 | 1 |
| Deep vein thrombosis | 0.4 | 0 | .240 |
| Renal insufficiency | 0 | 0 | - |
| Pulmonary embolism | 0.9 | 0 | .057 |
| Ventilator > 48 hours | 0 | 0.1 | 1 |
| Acute renal failure | 0 | 0 | - |
| CVA with neurologic deficit | 0.9 | 0 | .057 |
| Nerve injury | 0 | 0 | - |
| Cardiac arrest requiring CPR | 0 | 0 | - |
| Myocardial infarction | 0 | 0 | - |
| Sepsis | 0.4 | 0.4 | 1 |
| Septic shock | 0 | 0.1 | 1 |
| Mortality, % | 0.4 | 0 | - |
| Total operating time, mean minutes ± SD | 92.1 ± 59.9 | 53.6 ± 42.0 | < .001* |
| Length of stay, mean days ± SD | 1.0 ± 0.7 | 0.9 ± 1.5 | .605 |
| Reoperation, % | 3.4 | 1.6 | .110 |
| Readmission, % (n = 294) | 6.1 | 1.4 | .041* |

Abbreviations: CPR, cardiopulmonary resuscitation; CVA, cerebrovascular accident; SD, standard deviation; SSI, surgical site infection

* denotes statistical significant (P < 0.05)

**Table 4. Multivariate regression for complications in sleep apnea surgery by resident involvement.**

|  | Complication rate (%) | P value | Odds Ratio (95% Confidence Interval) |
|---|---|---|---|
| Surgical complications |  |  |  |
| With resident involvement | 1.3 | .945 | 1.06 (0.20–5.73) |
| Attending alone | 0.8 | Reference | Reference |
| Medical complications |  |  |  |
| With resident assistance | 2.1 | 0.799 | 1.20 (0.30–4.79) |
| Attending alone | 1.1 | Reference | Reference |
| Overall complications |  |  |  |
| With resident assistance | 3.4 | .794 | 1.16 (0.39–3.43) |
| Attending alone | 1.9 | Reference | Reference |

* denotes statistical significant (P < 0.05)

ACS NSQIP database to offer insights into this balance, particularly in the realm of head and neck OSA surgery.

In our study, 24.0% of cases were found to involve a resident, of which 48.7% involved a senior resident and 51.3% involved a junior resident. On multivariate analysis, resident involvement did not increase 30-day surgical, medical, or overall complication rates (Table 4). Similar findings have been corroborated by other studies across a wide range of fields [10, 12, 17–20, 23], all demonstrating the safety of resident involvement in surgical procedures. After categorization of involved residents into junior or senior resident cohorts based on PGY, we similarly found no increase in odds of surgical, medical, or overall complication rates (Table 7). Other studies have corroborated these findings as well, demonstrating no significantly increased odds of complications with increasing trainee PGY [15, 46–48].

Notably, our study did demonstrate increased rates of readmission with resident involvement (7.4% versus 1.3%, p = 0.009) in these surgical procedures. More specifically, rates of readmission were highest in cases involving a senior resident (8.3%) as compared to those involving a junior resident (6.8%) or those performed by the attending alone (1.3%, p = 0.017). However, it must be noted that only 294 (30.2%) of our cases provided any information pertaining to patient readmission status; therefore, the sample size may have affected the accuracy of group comparisons. Furthermore, the absence of detailed data pertaining to the cause for readmission prevents us from making any definitive conclusions. For example, postoperative hemorrhage as a cause for readmission is likely not adequately captured through proxies such as "requiring a blood transfusion within 72 hours," especially in patients experiencing postoperative bleeding not severe enough to warrant a blood transfusion or those who require a transfusion after the 72-hour mark. The increased readmission rates in cases with resident involvement may be attributed to the complexity and acuity of patients selected for these teaching cases. Larger tertiary care centers, often equipped with comprehensive teams including residents, may tackle more complex cases, inherently carrying a higher risk of readmission when compared to simpler surgeries performed by the attending alone in an outpatient surgical center. Furthermore, larger hospitals may preferentially include residents on more challenging cases to provide additional assistance. To address this, we employed multivariate regression to control for demographic and preoperative variables significant on univariate analysis. Conversely, outpatient cases tend to involve healthier patients, typically leading to decreased risks of postoperative complications and readmission in these cases [49]. In our study, the senior resident cohort demonstrated the largest, albeit statistically non-significant, proportion of inpatient cases (35.2%) compared to the junior resident cohort (31.1%) or the

**Table 5. Overall patient demographics and comorbidities in sleep apnea surgery by trainee postgraduate year.**

| | Attending alone N = 741 | Junior resident involvement N = 120 | Senior resident involvement N = 114 | P value |
|---|---|---|---|---|
| Demographics ± | | | | |
| Age, years ± SD | 43.7 ± 12.0 | 43.0 ± 12.5 | 45.4 ± 11.8 | .275 |
| Sex, % | | | | .197 |
| Female | 25.4 | 32.5 | 29.8 | |
| Male | 74.6 | 67.5 | 70.2 | |
| Race, % | | | | < .001* |
| Black | 5.7 | 20.0 | 18.4 | |
| Other | 3.9 | 2.5 | 2.6 | |
| Unknown | 22.4 | 23.3 | 21.1 | |
| White | 68.0 | 54.2 | 57.9 | |
| BMI, kg/m$^2$ ± SD | 32.9 ± 7.7 | 32.8 ± 6.4 | 32.1 ± 7.2 | .530 |
| Comorbidities, % | | | | |
| Diabetes | 10.4 | 6.7 | 9.6 | .445 |
| Current smoker | 18.5 | 19.2 | 14.9 | .624 |
| Dyspnea | 7.3 | 5.8 | 9.6 | .525 |
| Alcohol | 1.2 | 5.0 | 2.6 | .014* |
| Pneumonia | 0 | 0 | 0 | - |
| Weight loss | 0 | 0 | 0.9 | .023* |
| COPD | 0.7 | 3.3 | 2.6 | .018* |
| CHF | 0 | 0.8 | 0 | .028* |
| Ventilator dependent | 0.1 | 0 | 0 | .854 |
| Dialysis dependent | 0.1 | 0 | 0 | .854 |
| Previous MI | 0 | 0 | 0.9 | .023* |
| Previous PCI | 2.2 | 0 | 0.9 | .184 |
| Previous cardiac surgery | 1.5 | 0 | 1.8 | .386 |
| Angina | 0.1 | 0 | 2.6 | < .001* |
| Hypertension | 31.4 | 35.8 | 28.1 | .436 |
| TIA | 0.7 | 0 | 0.9 | .633 |
| CVA with neurologic deficit | 0.1 | 0 | 3.5 | < .001* |
| CVA without neurologic deficit | 0.8 | 0.8 | 2.6 | .195 |
| Disseminated cancer | 0 | 0 | 0 | - |
| Steroid use | 0.8 | 0 | 1.8 | .330 |
| Bleeding disorders | 0.4 | 0.8 | 0 | .608 |
| Systemic sepsis | 0.3 | 0.8 | 0 | .482 |
| Wound infection | 0.3 | 0 | 0 | .729 |
| Outpatient status | 71.6 | 68.9 | 64.8 | .328 |
| Functional status | | | | .608 |
| Dependent | 0.4 | 0.8 | 0 | |
| Independent | 99.6 | 99.2 | 100.0 | |
| ASA class | | | | .057 |
| 1 | 5.7 | 1.7 | 3.5 | |
| 2 | 59.5 | 69.2 | 54.9 | |
| 3 | 34.4 | 29.2 | 39.8 | |
| 4 | 0.4 | 0 | 1.8 | |

*(Continued)*

**Table 5.** (Continued)

| | Attending alone<br>N = 741 | Junior resident involvement<br>N = 120 | Senior resident involvement<br>N = 114 | P value |
|---|---|---|---|---|
| WRVU | 12.3 ± 6.2 | 12.6 ± 6.4 | 12.5 ± 5.5 | .913 |

Abbreviations: ASA, American Society of Anesthesiologists classification; BMI, body mass index; COPD, chronic obstructive pulmonary disease; CHF, congestive heart failure; CVA, cerebrovascular accident; MI, myocardial infarction; PCI, percutaneous coronary intervention; TIA, transient ischemic attack; SD, standard deviation; WRVU, work relative value unit

* denotes statistical significance (P < 0.05)

attending alone cohort (28.4%, p = 0.328), potentially supporting the explanation that residents are typically involved in more complex cases. In attempt to account for case complexity, we calculated the overall WRVU to act as a proxy for case complexity, as similarly calculated by Baker et al. in their analysis of resident involvement on postoperative outcomes in outpatient otolaryngology surgeries [12]. Nonetheless, we found no significant difference in overall

**Table 6. Postoperative complications in sleep apnea surgery by trainee postgraduate year.**

| | Attending alone<br>N = 741 | Junior resident involvement<br>N = 120 | Senior resident involvement<br>N = 114 | P value |
|---|---|---|---|---|
| Overall complications, % | 1.9 | 0.8 | 6.1 | .009* |
| Surgical complications overall, % | 0.8 | 0 | 2.6 | .088 |
| Superficial SSI | 0.4 | 0 | 0.9 | .576 |
| Deep SSI | 0 | 0 | 0 | - |
| Organ/space SSI | 0.3 | 0 | 0 | .729 |
| Wound disruption | 0.1 | 0 | 1.8 | .012* |
| Blood transfusion within 72 hours | 0 | 0 | 0 | - |
| Medical complications overall, % | 1.1 | 0.8 | 3.5 | .096 |
| Pneumonia | 0.4 | 0 | 0 | .622 |
| Unplanned reintubation | 0.7 | 0 | 0 | .452 |
| Urinary tract infection | 0.4 | 0 | 0 | .622 |
| Deep vein thrombosis | 0 | 0.8 | 0 | .028* |
| Renal insufficiency | 0 | 0 | 0 | - |
| Pulmonary embolism | 0 | 0.8 | 0.9 | .042* |
| Ventilator > 48 hours | 0.1 | 0 | 0 | .854 |
| Acute renal failure | 0 | 0 | 0 | - |
| CVA with neurologic deficit | 0 | 0 | 1.8 | .001* |
| Nerve injury | 0 | 0 | 0 | - |
| Cardiac arrest requiring CPR | 0 | 0 | 0 | - |
| Myocardial infarction | 0 | 0 | 0 | - |
| Sepsis | 0.4 | 0 | 0.9 | .576 |
| Septic shock | 0.1 | 0 | 0 | .854 |
| Mortality, % | 0 | 0 | 0.9 | - |
| Total operating time, mean minutes ± SD | 53.6 ± 42.0 | 94.0 ± 53.6 | 90.2 ± 66.1 | < .001* |
| Length of stay, mean days ± SD | 0.9 ± 1.5 | 1.0 ± 0.7 | 0.9 ± 0.8 | .914 |
| Reoperation, % | 1.6 | 2.5 | 4.4 | .142 |
| Readmission, % (n = 294) | 1.4 | 3.8 | 10.3 | .019* |

Abbreviations: CPR, cardiopulmonary resuscitation; CVA, cerebrovascular accident; SD, standard deviation; SSI, surgical site infection

* denotes statistical significant (P < 0.05)

**Table 7. Multivariate regression for complications in sleep apnea surgery by trainee postgraduate year.**

|  | Complication rate (%) | P value | Odds Ratio (95% Confidence Interval) |
|---|---|---|---|
| Surgical complications |  |  |  |
| Senior resident involvement | 2.6 | .429 | 2.01 (0.36–11.37) |
| Junior resident involvement | 0 | 1 | - |
| Attending alone | 0.8 | Reference | Reference |
| Medical complications |  |  |  |
| Senior resident involvement | 3.5 | .725 | 1.32 (0.28–6.12) |
| Junior resident involvement | 0.8 | .435 | 0.40 (0.04–4.07) |
| Attending alone | 1.1 | Reference | Reference |
| Overall complications |  |  |  |
| Senior resident involvement | 6.1 | .417 | 1.62 (0.51–5.20) |
| Junior resident involvement | 0.8 | .205 | 0.23 (0.02–2.20) |
| Attending alone | 1.9 | Reference | Reference |

* denotes statistical significant (P < 0.05)

WRVU among the different cohorts (junior resident: 12.6 ± 6.4, senior resident: 12.5 ± 5.5, attending alone: 12.3 ± 6.2, p = 0.913). However, the limitations of WRVU as a proxy for case complexity are well-documented, with substantial literature demonstrating its inability to effectively capture case complexity [50–54]. This discrepancy is partly because WRVU assignments are influenced by budgetary constraints. As detailed by Shah et al. in their analysis of the correlation between WRVU and surgical effort/complexity, the task of assigning WRVUs falls to a specialized group, the Relative Value Scale Update Committee (RUC), which operates within the confines of Medicare's budget limits, often necessitating reductions in WRVUs for certain services to accommodate increases in others, ultimately resulting in a WRVU assignment being subjective and vulnerable to outside influences [54]. As further demonstrated in Childers et al.'s study of WRVU and work measure, the field of otolaryngology, with 35 procedures analyzed, demonstrated that 80% of procedures had lower-than-expected WRVU, suggesting that WRVU, especially within otolaryngology, may not be a suitable proxy for case complexity [52]. The inaccuracy of overall WRVU as a proxy for case complexity highlights a limitation of the ACS NSQIP dataset. Given these considerations, further research is necessary to better elucidate if resident involvement truly impacts readmission rates in patients undergoing OSA surgery. In their analysis of the ACS NSQIP database on outpatient otolaryngology surgeries, Baker et al. found that resident involvement did not increase risk of readmission [12]. Likewise, both Brady et al. and Wong et al. found that rates of readmission did not significantly increase with resident involvement in head and neck free flap surgeries or endoscopic transsphenoidal pituitary surgery, respectively [15, 16]. These findings suggest that factors other than resident involvement may contribute to the observed differences seen between our cohorts.

Finally, in our study, we observed that resident involvement significantly prolonged the total operation time. Specifically, operations involving junior residents (94.0 ± 53.6 minutes) and senior residents (90.2 ± 66.1 minutes) both demonstrated significantly prolonged operative times when compared to operations performed by attending alone (53.6 ± 42.0 minutes). Although patient safety was not compromised, the efficiency of the procedure was, as indicated by the almost doubled operative time in cases involving residents. This extended operative time may reflect the time spent teaching and learning within the operating room, as well as the decreased operative efficiency when a resident is the primary surgeon [55]. Several studies have demonstrated resident involvement to increase average operation time [12, 15, 19, 20,

56]. Moreover, growing literature suggests that stable surgical teams can notably decrease the duration of surgeries [56–58]. In contrast, at tertiary academic hospitals, where there is a regular rotation of surgical and anesthesia residents over weeks or months, can result in frequent changes within surgical teams. Such inconsistencies may further contribute to longer operative times. Moreover, there is a well-documented risk between extended operative time and heightened risk of postoperative complications [19, 59–61]. In addition to clinical implications, prolonged operative time also has significant economic impacts. Intuitively, longer surgical duration inherently increases costs for healthcare facilities by extending the use of the operating room, raising staff wages and facility overhead for each procedure, and reducing the number of surgeries able to be performed per day. Research, including that by Vieira et al. examining resident participation in otolaryngology surgeries across over two million patients, supports the notion that increased operative times raise overall surgical costs and hospital strain. This extended surgical duration has also been linked to a higher incidence of certain postoperative complications, such as superficial surgical site infections [19]. These findings underscore the importance of finding a balance between providing thorough resident training and minimizing the impact on healthcare resources and patient outcomes.

Ultimately, our analysis of the ACS NSQIP database demonstrates that resident involvement in head and neck OSA surgery does not negatively impact 30-day postoperative surgical, medical, or overall outcomes. However, resident involvement did significantly increase the incidence of readmission and almost doubled the average operative time. Further research is needed to explore the long-term outcomes of surgeries with resident involvement and to identify strategies to optimize both educational and patient care goals in surgical training.

Our study is not without limitations. The ACS NSQIP dataset, drawing data from a variety of hospital systems around the world, may not be fully representative of all hospital centers or accurately reflect the influence of resident involvement or their level of postgraduate year on postoperative outcomes in OSA surgery. Furthermore, ACS NSQIP only collects data up to 30 days postoperatively, restricting our ability to assess outcomes that occur beyond this period. Additionally, the dataset only identifies the highest postgraduate year among residents involved, failing to specify if additional residents were involved in the surgical procedure and, if so, what their training PGY was. Similarly, the specialties of participating trainees are not reported, which could lead to variations in postoperative outcomes depending on their residency training model. Likewise, ACS NSQIP does not clarify the specific roles or the extent of involvement of residents in the surgeries, possibly influencing our results. Trainees of higher postgraduate years with more experience in sleep apnea surgery may have played a larger role in these surgeries versus those involving junior level residents where the attending surgeon may have performed the majority of the procedure. Importantly, the ACS NSQIP dataset ceased to report resident involvement in surgeries after the year 2012. This omission could potentially restrict our study's capacity to evaluate the evolution of surgical treatment for OSA beyond these years. Although the use of overall WRVUs as a proxy for case complexity has been utilized in previous NSQIP studies [12], overall WRVU may not completely account for varying case complexities as previously discussed.

## Conclusion

Resident involvement in head and neck OSA surgery did not significantly increase the odds of surgical, medical, or overall postoperative complications. Similarly, the PGY of assisting residents was not found to increase these odds. However, resident involvement was associated with significantly increased incidence of readmission and almost doubled the average operation time. These findings suggest that teaching hospitals have, thus far, successfully balanced

training future surgeons with the optimization of patient safety in head and neck OSA surgery, but that further attention should be placed on improving surgical efficiency to reduce operative time.

## Author Contributions

**Conceptualization:** Mihai A. Bentan, Ryan Nord.

**Data curation:** Mihai A. Bentan, Elizabeth Mastoloni, Lawrance Lee, Ryan Nord.

**Formal analysis:** Mihai A. Bentan, Elizabeth Mastoloni, Lawrance Lee, Ryan Nord.

**Methodology:** Mihai A. Bentan, Elizabeth Mastoloni, Lawrance Lee, Ryan Nord.

**Project administration:** Ryan Nord.

**Supervision:** Mihai A. Bentan, Lawrance Lee.

**Writing – original draft:** Mihai A. Bentan, Elizabeth Mastoloni, Lawrance Lee, Ryan Nord.

**Writing – review & editing:** Mihai A. Bentan, Elizabeth Mastoloni, Lawrance Lee, Ryan Nord.

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
