## [Decision Letter · Decision Letter 0]

4 Oct 2024

PONE-D-24-20350The impact of resident involvement and postgraduate year on head and neck surgery for obstructive sleep apneaPLOS ONE

Dear Dr. Bentan,

Thank you for submitting your manuscript to PLOS ONE. After careful consideration, we feel that it has merit but does not fully meet PLOS ONE’s publication criteria as it currently stands. Therefore, we invite you to submit a revised version of the manuscript that addresses the points raised during the review process.

We look forward to receiving your revised manuscript.

Kind regards,

Alessandro Cannavo

Academic Editor

PLOS ONE

**Journal Requirements:**

2. Thank you for stating the following in the Competing Interests: 

I have read the journal's policy and the authors of this manuscript have the following competing interests: Dr. Ryan Nord is a consultant for Inspire Medical, Nyxoah, and SIESTA medical. All other authors have declared that no competing interests exist.    

We note that one or more of the authors are employed by a commercial company.

3. In the online submission form, you indicated that The data is part of the ACS-NSQIP PUF database maintained by the American College of Surgeons. The data is available free of charge from the American College of Surgeons for members of hospitals participating in the NSQIP. Data access queries for non-members can be directed to the Corresponding Author, Mihai Bentan (bentanm@vcu.edu)

Reviewers' comments:

Reviewer's Responses to Questions

**Comments to the Author**

1. Is the manuscript technically sound, and do the data support the conclusions?

Reviewer #1: Yes

2. Has the statistical analysis been performed appropriately and rigorously? 

Reviewer #1: Yes

3. Have the authors made all data underlying the findings in their manuscript fully available?

Reviewer #1: Yes

4. Is the manuscript presented in an intelligible fashion and written in standard English?

Reviewer #1: Yes

5. Review Comments to the Author

**Reviewer #1:** Really appreciate the work of all authors on the manuscript. In my opinion, the results inside may help surgeons clarify whether resident involvement induces higher rate of sleep surgical events postoperatively. Personal consideration as below could make this manusciript more comprehensive:

Authors advocated some opinions, trying to address the analyzed result that increased incidence of readmission occurred in the resident group. Based on personal experience, readmission after upper airway surgery was usually attributed to postoperative bleeding that required emergent check bleeding or closely observation. But even in such cases, blood transfusion was not a routine procedure unless critical vital sign or extremely low Hgb level. In the manuscript, only the parameter "Blood transfusion within 72 hours" was included. Hence, the surgical complication about postoperative bleeding that required readmission might be partially ignored. So...is it available to introduce the event of postoperative bleeding that requires check bleeding or closely monitor as a factor, to compare the surgical complication rate between residents and attending group ? If not, please add the description in Discussion or limitation paragraph, to illustrate "Blood transfusion within 72 hours" is used to represent the complication of postoperative bleeding rather than other parameters, for example, the event of check bleeding surgery.

6. PLOS authors have the option to publish the peer review history of their article (what does this mean?). If published, this will include your full peer review and any attached files.

Reviewer #1: No

---

## [Author Response · Author response to Decision Letter 0]

19 Nov 2024

Dear Editors and Reviewers,

Thank you for inviting our team to submit a revised draft of our manuscript entitled, “The impact of resident involvement and postgraduate year on head and neck surgery for obstructive sleep apnea” (PONE-D-24-20350) to PLOS ONE. We appreciate the effort and time that the reviewers dedicated in providing their insight and helpful feedback. We agree that this extra effort further strengthens our manuscript. We have incorporated the suggestions from the incite that was provided by the reviewers. We anticipate that the edits and responses that we have shared below addresses the concerns and recommendations you and the reviewers have provided.

To aid in the facilitation of your review of the revisions, the following is a point-by-point response to the questions and comments delivered in your revision letter. To facilitate readthrough, we have separated each question/comment using the header COMMENT and then replied to each comment via the header RESPONSE. 

Thank you once again for your time and consideration.

COMMENT: We note that your "Manuscript and Cover Letter" files are duplicated on your submission. Please remove any unnecessary or old files from your revision, and make sure that only those relevant to the current version of the manuscript are included.

RESPONSE: All unnecessary / old files have been removed from the submission. Only the current versions of the manuscript and cover letter have been included. We apologize for any confusion caused by this action.

COMMENT: We understand that you have updated your data availability statement to read as follows: "The data is part of the ACS-NSQIP PUF database maintained by the American College of Surgeons. The data is available free of charge from the American College of Surgeons for members of hospitals participating in the NSQIP. Data access queries for non-members can be directed to the Corresponding Author, Mihai Bentan (bentanm@vcu.edu).

Public disposition of this data would breach the Data Use Agreement (and therefore copyright laws) as imposed by the American College of Surgeons, as detailed in the Participant Use Data File ‘Access to Chapter 4’ section: “Chapter 4 (for both Essential and Procedure Targeted data) contains proprietary ACS NSQIP programmatic information. Your use of Chapter 4 is restricted to clarifying definitions for your PUF-based research purposes. The contents of Chapter 4 shall not be shared, quoted, distributed, or disseminated in any published papers, reports, manuscripts, etc. Limited paraphrasing of Chapter 4 definitions, to provide methodological clarity in research publications, is permissible. Any use or redistribution of Chapter 4 for commercial purposes is strictly prohibited.” Therefore, given these copyright laws, we are unable to provide the data underlying our finals in any of the aforementioned methods (a-c) so as to avoid violating copyright laws. However,

individuals/readers interested seeking permission to use the data may follow contact the ACS who will process a request, as detailed in the following: “Permission to use and disclose the PUF is granted from the ACS NSQIP to each Data Recipient. The Data Recipient must be an employee of the participating Hospital. Intentional misrepresentation of employment at the Hospital by the Data Recipient will void this Agreement, prohibit use of the PUF by the Data Recipient, and result in legal action taken by the ACS NSQIP. The ACS NSQIP also reserves the right to deny access to the PUF at its discretion.” Further information regarding all of this information can be found at the following link: https://www.facs.org/media/ezaj2g5p/datauseagreementacsnsqippuf.pdf

The data is owned by a third-part institution (ACS NSQIP) and therefore they cannot be shared by any of the authors of this study. The following URL: https://www.facs.org/quality-programs/data-and-registries/pediatric/participant-use-data-file/participant-use-request-form/ should be utilized to identify the appropriate non-author contact for the location/state in which the requesting individual resides. Other individuals are able to access this data in the same manner that these authors have. We report that these authors did not have any specific access privileges that others would not have had."

However, this statement does not provide non-author contact information for requesting these data. Please include a non-author email address in this statement.

RESPONSE: Thank you for bringing this to our attention. The following non-author email address, acsnsqip@facs.org, may be utilized for data requests. This information has been added to the data availability statement, which now reads as follows:

The data is part of the ACS-NSQIP PUF database maintained by the American College of Surgeons. The data is available free of charge from the American College of Surgeons for members of hospitals participating in the NSQIP. Data access queries for non-members can be directed to the Corresponding Author, Mihai Bentan (bentanm@vcu.edu).

Public disposition of this data would breach the Data Use Agreement (and therefore copyright laws) as imposed by the American College of Surgeons, as detailed in the Participant Use Data File ‘Access to Chapter 4’ section: “Chapter 4 (for both Essential and Procedure Targeted data) contains proprietary ACS NSQIP programmatic information. Your use of Chapter 4 is restricted to clarifying definitions for your PUF-based research purposes. The contents of Chapter 4 shall not be shared, quoted, distributed, or disseminated in any published papers, reports, manuscripts, etc. Limited paraphrasing of Chapter 4 definitions, to provide methodological clarity in research publications, is permissible. Any use or redistribution of Chapter 4 for commercial purposes is strictly prohibited.” Therefore, given these copyright laws, we are unable to provide the data underlying our finals in any of the aforementioned methods (a-c) so as to avoid violating copyright laws. However,

individuals/readers interested seeking permission to use the data may follow contact the ACS who will process a request, as detailed in the following: “Permission to use and disclose the PUF is granted from the ACS NSQIP to each Data Recipient. The Data Recipient must be an employee of the participating Hospital. Intentional misrepresentation of employment at the Hospital by the Data Recipient will void this Agreement, prohibit use of the PUF by the Data Recipient, and result in legal action taken by the ACS NSQIP. The ACS NSQIP also reserves the right to deny access to the PUF at its discretion.” Further information regarding all of this information can be found at the following link: https://www.facs.org/media/ezaj2g5p/datauseagreementacsnsqippuf.pdf

The data is owned by a third-part institution (ACS NSQIP) and therefore they cannot be shared by any of the authors of this study. The following URL: https://www.facs.org/quality-programs/data-and-registries/pediatric/participant-use-data-file/participant-use-request-form/ should be utilized to identify the appropriate non-author contact for the location/state in which the requesting individual resides. Other individuals are able to access this data in the same manner that these authors have. We report that these authors did not have any specific access privileges that others would not have had. Data requests may be performed using the aforementioned URL link or via the following email address: acsnsqip@facs.org.

---

## [Decision Letter · Decision Letter 1]

27 Dec 2024

The impact of resident involvement and postgraduate year on head and neck surgery for obstructive sleep apnea

PONE-D-24-20350R1

Dear Dr. Bentan,

We’re pleased to inform you that your manuscript has been judged scientifically suitable for publication and will be formally accepted for publication once it meets all outstanding technical requirements.

Kind regards,

Alessandro Cannavo

Academic Editor

PLOS ONE

Additional Editor Comments (optional):

Reviewers' comments:

Reviewer's Responses to Questions

**Comments to the Author**

1. If the authors have adequately addressed your comments raised in a previous round of review and you feel that this manuscript is now acceptable for publication, you may indicate that here to bypass the “Comments to the Author” section, enter your conflict of interest statement in the “Confidential to Editor” section, and submit your "Accept" recommendation.

Reviewer #1: All comments have been addressed

2. Is the manuscript technically sound, and do the data support the conclusions?

Reviewer #1: Yes

3. Has the statistical analysis been performed appropriately and rigorously? 

Reviewer #1: N/A

4. Have the authors made all data underlying the findings in their manuscript fully available?

Reviewer #1: No

5. Is the manuscript presented in an intelligible fashion and written in standard English?

Reviewer #1: Yes

6. Review Comments to the Author

Reviewer #1: Thank you for the work on the revised manuscript. The description concerning postoperative hemorrhage not fully captured was added. I have no other comments.

7. PLOS authors have the option to publish the peer review history of their article (what does this mean?). If published, this will include your full peer review and any attached files.

Reviewer #1: No

---

## [Editor Report · Acceptance letter]

2 Jan 2025

PONE-D-24-20350R1 

PLOS ONE

Dear Dr. Bentan, 

I'm pleased to inform you that your manuscript has been deemed suitable for publication in PLOS ONE. Congratulations! Your manuscript is now being handed over to our production team.

Kind regards, 

on behalf of

Dr. Alessandro Cannavo 

Academic Editor

PLOS ONE